# OASIS: An Optimized Approach to Systematic Calibration Data Selection

## Abstract

Post-training pruning is a critical technique for compressing Large Language Models. However, as shown in previous research, its effectiveness is highly sensitive to the small set of calibration data used for estimating parameter importance. Current calibration data selection relies on simple heuristics like random sampling or entropy, which often leads to suboptimal and inconsistent pruning outcomes: the same pruning method applied with different calibration data can cause up to 3× variance in post-pruning perplexity. In this work, we reveal the source of this inconsistency: calibration samples are not equally important; a quality hierarchy exists within any data pool. Not only does mixing high- and low-quality data cause a performance degradation, but the quality of the sample is context-dependent, changing with the specific model and pruning algorithm, rendering static filtering infeasible and necessitating an adaptive solution. Therefore, we introduce OASIS, the first end-to-end framework that directly optimizes calibration data selection with respect to the pruned model's downstream performance. OASIS leverages a differentiable soft-mask proxy to propagate task-level gradients back to the calibration data, enabling dynamic discovery of the most beneficial subset. Experiments show that our approach improves the performance of diverse state-of-the-art pruning methods, establishing a new standard for data-aware model compression.

## 1 Introduction

Large Language Models (LLMs) based on the Transformer architecture (Vaswani et al., 2017) have achieved state-of-the-art performance on a wide array of natural language tasks. However, their ever-increasing size and computational demands present a significant barrier to widespread deployment and research, limiting access to those with substantial hardware resources. Consequently, model compression techniques, particularly pruning, have become essential for creating more efficient and accessible models (Frantar and Alistarh, 2023; An et al., 2024; Ma et al., 2023; Xia et al., 2024a).

However, the efficacy of most pruning methods, both structured and unstructured, critically depends on a small set of calibration data. This data is used to estimate parameter importance saliency scores that guide the pruning decisions. A growing body of evidence shows that the performance of the final pruned model is highly sensitive to the choice of this calibration data (Ji et al., 2024; Kurz et al., 2024; Williams and Aletras, 2024a), with factors like quality and diversity having an outsized impact on the outcome (Ai et al., 2025; Bandari et al., 2024).

Despite its critical importance, the selection of calibration data is often treated as an afterthought. Common practice relies on simple heuristics, such as randomly sampling a few dozen examples from a large web corpus like C4 or Wikipedia (Sun et al., 2023;

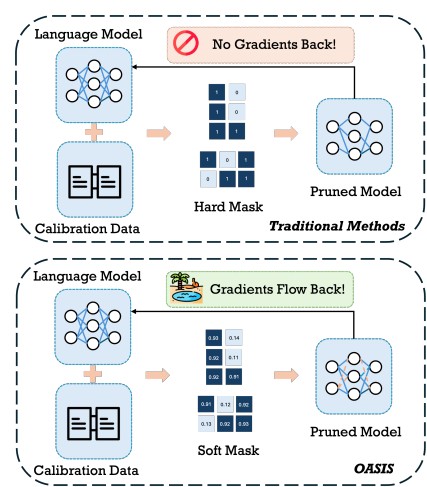

Figure 1: **Traditional vs. OASIS.** Traditional pruning blocks gradients, while **OASIS** enables end-to-end feedback via soft masks and learnable calibration importance weights.

Frantar and Alistarh, 2023; Bandari et al., 2024). In this work, we first conduct a fine-grained analysis that reveals why such heuristic approaches are suboptimal. Our investigation demonstrates that calibration samples possess a clear quality hierarchy ("golden," "mediocre," and "detrimental" samples). We find that while combining high-quality samples yields synergistic benefits, mixing them across different quality tiers causes a poisoning effect. A single low-quality ("detrimental") sample can contaminate the entire set and severely degrade the performance of a high-quality ("golden") one. Furthermore, our analysis uncovers a deeper challenge: the quality of a given sample is not a universal property: the set of "golden" samples varies significantly with the specific pruning method and model being used. This dependency means that a simple, static approach to "mine" for universally effective data is infeasible, which presents a significant technical challenge and motivates the need for an adaptive mechanism that can select the optimal calibration set for each unique pruning scenario.

The goal for this data selection problem is connecting the choice of calibration data to the pruned model's performance. This connection is broken by the non-differentiable nature of the hard, binary mask used in pruning, which prevents end-to-end gradient flow (Figure 1). Our core contribution is to introduce a novel, end-to-end training paradigm that directly solves this problem: we reframe the selection process as a learnable optimization task. By leveraging a differentiable soft-mask proxy, we create a *fully differentiable* pipeline that allows gradients from the final task performance to inform which calibration samples are most valuable. This allows us to directly learn an optimal data subset that maximizes the effectiveness of any underlying pruning algorithm, replacing arbitrary heuristics with a principled, performance-driven approach.

In this work, we introduce **OASIS**, an **O**ptimized **A**pproach to **S**ystematic cal**I**bration data **S**election. Our primary contributions are:

- A thorough fine-grained analysis that systematically demonstrates the synergistic and degradative effects of individual calibration data points, exposing the critical flaws in common heuristic-based selection methods.
- The first method to directly optimize the selection of calibration data based on the final pruned model's performance through a gradient-based framework with soft-pruning proxy, moving beyond metrics like entropy or simple random sampling.
- Comprehensive experiments showing that our method significantly improves the performance of multiple state-of-the-art structured and unstructured pruning algorithms on various LLMs (Llama3 Family, Qwen2.5), establishing a new, robust baseline for calibration data selection.

## 2 RELATED WORK

**LLM Pruning.** Pruning aims to reduce LLM size and accelerate inference by removing redundant parameters, and can be categorized into three classes. *Unstructured pruning* (e.g., SparseGPT (Frantar and Alistarh, 2023), Wanda (Sun et al., 2023)) eliminates individual weights but yields irregular patterns with limited hardware benefit. *Semi-structured pruning* enforces patterns such as 2:4 sparsity (Fang et al., 2024; Zheng et al., 2024), offering hardware friendliness but requiring specialized support. *Structured pruning*, the most practical in deployment, removes entire neurons, heads, or layers for direct speedups. Early works (LLM-Pruner (Ma et al., 2023), Sheared Llama (Xia et al., 2024a), SlimGPT (Ling et al., 2024)) mainly used local saliency, while recent methods (Adapt-Pruner (Wang et al., 2025), FLAP (An et al., 2024), ShortGPT (Men et al., 2024), NIRVANA (Ai et al., 2025)) adopt global, layer-wise strategies. Other innovations include PCA-based designs (SliceGPT (Ashkboos et al., 2024), Olica (He and Lin, 2025)). Despite progress, *all approaches remain highly sensitive to calibration data*, making data selection important for pruning effectiveness.

**Calibration Data in Pruning.** The efficacy of most post-training pruning methods hinges on a small set of *calibration data* used to estimate activation or gradient statistics for importance scoring. However, a growing body of work demonstrates that pruning outcomes are highly sensitive to the properties of this data. The selection process is critical, with factors like quality, diversity, and alignment with the model's pretraining distribution significantly influencing which components are preserved and the final model performance (Williams and Aletras, 2024b; JAISWAL et al., 2024; Bandari et al., 2024). To address this sensitivity, several research directions have emerged. One approach is to generate synthetic calibration data directly from the language model itself, aiming to create samples that better reflect its internal knowledge distribution (Williams and Aletras, 2024a; Ji et al., 2024). Yet, the underlying mechanisms of why certain data distributions lead to better pruning

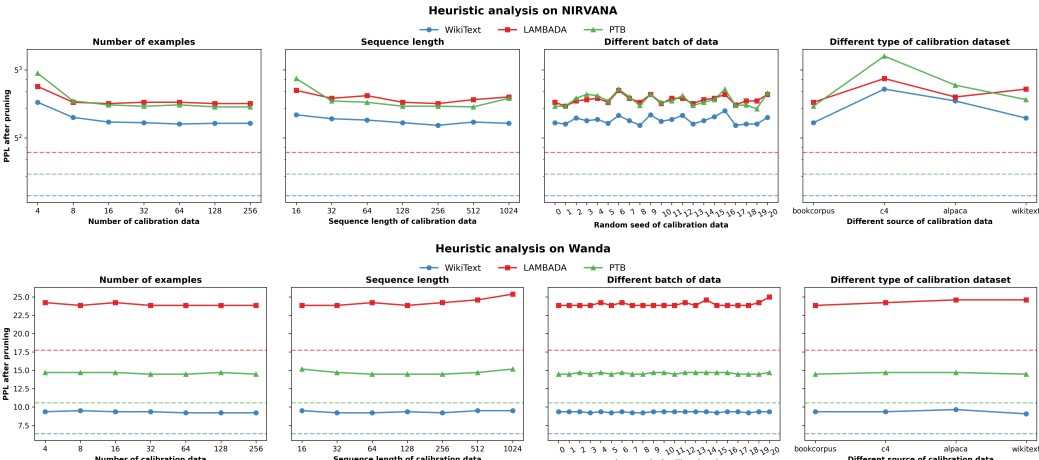

Figure 2: A **macro-level** analysis of how different heuristic choices for calibration data impact pruning performance. The top and bottom rows show results for the structured pruner (NIRVANA) and the unstructured pruner (Wanda), respectively. Each column investigates a different variable: (from left to right) the number of samples, sequence length, the random data batch, and the data source. The y-axis represents the post-pruning perplexity on three evaluation datasets: WikiText, PTB, and Lambda, indicated by separate colored lines. Dashed lines show the baseline perplexity of the unpruned model for comparison.

outcomes remain underexplored. This data sensitivity is particularly pronounced in *structured pruning* (Ai et al., 2025), where aggregating saliency scores over large components amplifies biases from the calibration data, leading to suboptimal decisions. Reflecting this challenge, specialized research has focused on curating calibration data for specific goals, such as language-specific pruning (Kurz et al., 2024; Zeng et al., 2024) and task-specific customization (Zhao et al., 2025), highlighting the need for data that aligns closely with the desired post-pruning model behavior.

**Data Selection.** Data selection is critical across all aspects of large language model training (Albalak et al., 2024), including pre-training (Gu et al., 2024), fine-tuning (Kang et al., 2024), instruction tuning (Xia et al., 2024b), and in-context learning (Zhang et al., 2022). Beyond accelerating the training process, data selection can significantly enhance model performance using a core subset of examples. Foundational data selection methods often employ a rank-and-select strategy, prioritizing the top-$k$ data points based on various statistics such as training dynamics (Paul et al., 2021), entropy-based confidence and uncertainty (Kremer et al., 2014), and marginal gain derived from submodular functions (Wei et al., 2015; Bhatt et al., 2024). Besides these heuristic methods, recent advancements have shifted towards model-aware selection. These methods estimate the influence of data on the final model's performance by attributing a contribution score to each training example. Prominent methods in this area include TracIn (Pruthi et al., 2020; Xia et al., 2024b), influence functions (Koh and Liang, 2017; Kwon et al., 2023), and training path unrolling (Bae et al., 2024).

## 3 WHY HEURISTICS FAIL: MULTI-SCALE ANALYSIS OF CALIBRATION DATA

In this section, we explore the nuances of calibration data selection to better understand the factors driving pruning performance. While existing heuristic approaches provide a valuable foundation, their analysis can be inconsistent under different pruning methods. For example, a zero-order method like Wanda (Sun et al., 2023), with a saliency score of $S = \|\mathbf{W}\mathbf{X}^2\|$, relies on stable activation magnitudes and thus tends to favor data of high linguistic quality. In contrast, higher-order methods like SparseGPT (Frantar and Alistarh, 2023), which use second-order Hessian information $S = \|\mathbf{W}\|^2/\text{diag}(\mathbf{H})$, are sensitive to the complex curvature of the loss landscape. Consequently, their data preference is often non-intuitive, as the data that best reveals this curvature may not align with human judgments of text quality, as detailed in Appendix B and Appendix C. Furthermore, these effects may differ substantially between structured and unstructured pruning methods, given the higher sensitivity in the structured pruning method (Ai et al., 2025). To investigate the source of this variance and build a more robust selection strategy, our analysis proceeds in two stages: We

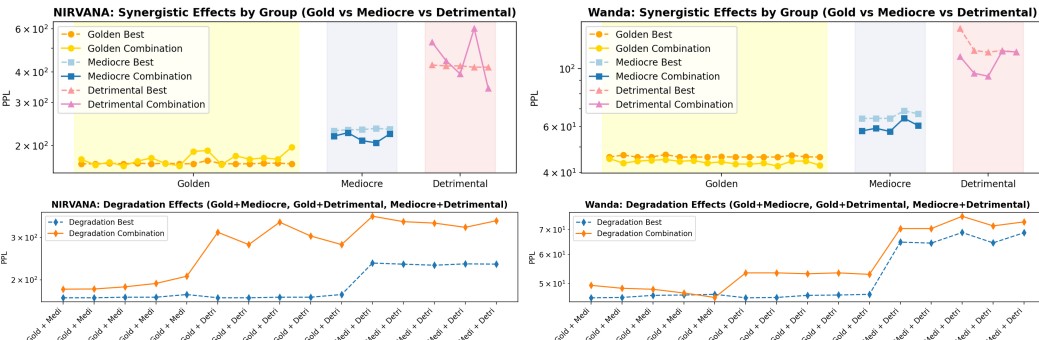

Figure 3: A **micro-level** analysis of calibration data interactions in NIRVANA (left) and Wanda (right). The top row investigates intra-tier synergy, comparing the performance of combining two samples from the same quality tier (e.g., "Golden Combination") against the performance of the better of the two individual samples ("Golden Best"). The bottom row demonstrates inter-tier degradation, showing the performance of combining samples from different tiers (e.g., "Golden + Detrimental") compared to the performance of the higher-quality sample in the pair ("Degradation Best").

begin with a (1) macro-level investigation to identify which data properties (e.g., number of data) have the most significant impact on performance, following previous work (Ji et al., 2024; Williams and Aletras, 2024a; Bandari et al., 2024). We then conduct a (2) micro-level analysis to uncover the underlying principles that explain these high-level observations.

### 3.1 MACRO-LEVEL ANALYSIS: DATA SOURCE OUTWEIGHS DATA QUANTITY

We begin by replicating and extending prior analyses, examining the impact of general calibration data properties such as quantity, sequence length, and source, isolating each factor in turn. Figure 2 illustrates these effects on both a stable unstructured method (Wanda, bottom half) and a highly sensitive structured method (NIRVANA, top half).

A common conclusion in previous calibration data selection methods is that "more data is better." Our results challenge this notion. As shown in Figure 2 (left), performance stays the same after 32 samples, and even becomes worse when increasing the sequence length. This indicates that simply adding more randomly chosen data is not an effective strategy for improving pruning outcomes.

In contrast, our analysis reveals that the **data source** is the most dominant performance factor, especially for structured pruning. As shown in Figure 2 (right), NIRVANA's perplexity fluctuates dramatically based on the data source (rightmost plot) and the specific random batch (third plot from left), while Wanda's performance is comparably unaffected by these choices. This macro-level analysis leads to a crucial insight: the failure of heuristic methods stems not from using too little data, but from failing to account for the immense difference in quality between data sources and even between random batches, which raises a more fundamental question: *what is happening at the individual sample level that causes these dramatic differences?*

### 3.2 MICRO-LEVEL ANALYSIS: THE PRINCIPLES OF SYNERGY AND DEGRADATION

To understand the mechanism behind the source-level effects observed previously, we conduct a novel fine-grained analysis to investigate the interactions between individual data samples. We first establish a baseline by evaluating each sample from a candidate pool individually, categorizing them into quality tiers ("Golden,""Mediocre," "Detrimental") based on the post-pruning perplexity. We then systematically combine pairs of samples to observe their joint effect, as shown in Figure 3. Our analysis reveals two fundamental principles that govern these interactions:

**Principle 1: Intra-Tier Synergy.** Our first key finding is that combining multiple samples from the same quality tier often produces a synergistic effect, leading to a better outcome than using the constituent samples alone, as demonstrated in Figure 3 (top). The lines representing "Golden + Golden" and "Mediocre + Mediocre" combinations show a lower perplexity than the dashed lines, which indicate the best-performing individual sample within each pair. This suggests that even high-quality samples contain diverse, complementary signals. Their combination creates a more

robust and comprehensive calibration set, allowing the pruning algorithm to make more informed decisions. Even combining two "detrimental" samples often results in a "less detrimental" outcome, indicating that diversity can mitigate some harm from low-quality data.

**Principle 2: Inter-Tier Degradation.** Conversely, our most critical finding is the principle of inter-tier degradation: when samples from different quality tiers are mixed, the lower-quality sample consistently "poisons" the higher-quality one, degrading the final performance. Figure 3 (bottom) starkly illustrates this phenomenon. The bars for "Golden + Detrimental" combinations show a dramatic increase in perplexity, far exceeding the performance of the "golden" sample when used alone (represented by the dashed line). The severity of this degradation is proportional to the quality gap; "Golden + Mediocre" combinations show a less severe, but still noticeable, performance drop.

These micro-level findings expose the fundamental flaw of heuristic methods like random sampling. A single "detrimental" sample, if randomly included in a calibration set, can effectively sabotage the benefits of several "golden" samples, leading to highly unpredictable and suboptimal pruning results. This leads to a clear conclusion: the ultimate goal of an effective selection strategy must be to construct a calibration set composed purely of "golden" samples, thereby maximizing synergy while completely avoiding the poisoning effect.

However, as our analysis in Appendix B reveals, a deeper challenge exists: the definition of a "golden" sample is not universal but is highly context-dependent, varying with the specific pruning method and model architecture. Therefore, the true challenge is not simply to filter a static list of "good" data. It is to create a dynamic and adaptive mechanism that can, for each unique pruning scenario, discover what constitutes a "golden" sample and then select a pure subset of them. This is a task that heuristics are fundamentally incapable of performing, which necessitates the new, learnable paradigm we introduce with OASIS.

---

**Takeaway: Why Heuristics Fail in Calibration Data Selection?**

Our multi-scale analysis reveals that the failure of heuristic calibration-data selection is **systemic, not incidental**: (1) simply adding more data or longer sequences yields no gain and can even degrade pruning quality; (2) calibration results are dominated by data **source**, showing that "random" heuristics are inherently unstable; (3) at the sample level, mixing quality tiers triggers a strong **poisoning effect**, where a single detrimental sample can nullify the benefit of multiple golden ones. Crucially, what counts as "golden" is **model- and method-dependent**, meaning that no static data can universally work. Therefore, a dynamic, learnable mechanism is **necessary** to adaptively discover and select the right calibration subset for each pruning scenario.

---

## 4 PROPOSED METHOD

In this section, we provide the preliminary and the details of our proposed method, OASIS. An illustrative diagram is shown in Figure 4.

### 4.1 PRELIMINARY AND NOTATION

Let $\mathbf{w} = (w_1, \ldots, w_d)^\top \in \mathbb{R}^d$ denote the full set of model parameters. Consider a calibration dataset $\mathcal{D}_{\text{cal}} = \{(x_i, y_i)\}_{i=1}^N$ together with a weight vector

$$\mathbf{u} = (u_1, \ldots, u_N)^\top, \quad u_i \geq 0, \quad \sum_{i=1}^N u_i = 1,$$

where each $u_i$ specifies the *importance* of sample $(x_i, y_i)$ to the calibration procedure. The calibration loss is then defined as $L_{\text{cal}}(\mathbf{w}; \mathbf{u}) = \sum_{i=1}^N u_i \, \ell\big(f(x_i; \mathbf{w}), y_i\big)$, where $\ell(\cdot, \cdot)$ is the per-sample loss function and $f(\mathbf{x}; \mathbf{w})$ denotes the model output. For each parameter $w_j$, let

$$S(w_j) = \text{Saliency}\big(w_j \,; L_{\text{cal}}(w; u)\big), \quad j = 1, \ldots, d,$$

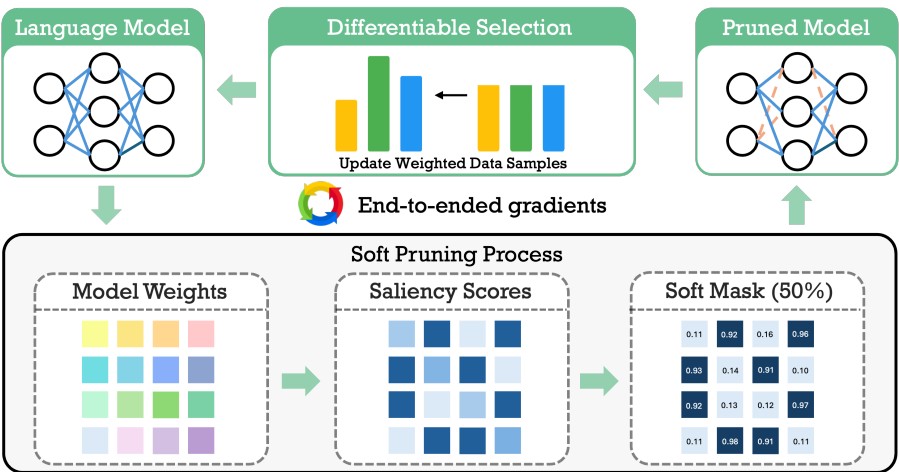

Figure 4: An overview of the OASIS framework for learning calibration data importance weights. OASIS reframes data selection as an end-to-end differentiable optimization loop.

denote its saliency score[1] w.r.t. the calibration loss. A binary mask is obtained via top-k thresholding:

$$m_j = \begin{cases} 1, & S(w_j \geq \text{Top}_k(S), \\ 0, & S(w_j < \text{Top}_k(S), \end{cases}$$

The pruned parameters $w'_j$ are then obtained by applying the hard mask $w'_j = m_j w_j$. Finally, given a downstream dataset $\mathcal{D}_{\text{task}}$, the *task loss* of the pruned model is

$$L_{\text{task}}(\mathbf{w}') = \sum_{(x,y) \in \mathcal{D}_{\text{task}}} \ell\big(f(x; \mathbf{w}'), y\big).$$

## 4.2 GRADIENT OF THE TASK LOSS W.R.T. CALIBRATION WEIGHTS

We investigate whether the downstream loss yields a nonzero gradient with respect to the calibration weights $u_i$. By the chain rule,

$$\frac{\partial L_{\text{task}}}{\partial u_i} = \sum_{j=1}^{d} \frac{\partial L_{\text{task}}}{\partial w'_j} \underbrace{\frac{\partial w'_j}{\partial m_j}}_{=w_j} \underbrace{\frac{\partial m_j}{\partial S_j}}_{=0 \text{ (a.e.)}} \underbrace{\frac{\partial S_j}{\partial u_i}}_{\neq 0 \text{ in general}}.$$

Since $m_j = \mathbf{1}[S_j \geq \text{Top}_k(S)]$, the term $\frac{\partial m_j}{\partial S_j}$ vanishes almost everywhere. Thus $\frac{\partial L_{\text{task}}}{\partial u_i} = 0$. Hence, hard pruning renders the gradient with respect to $\mathbf{u}$ non-differentiable, precluding direct optimization.

## 4.3 OASIS: PRUNING WITH SOFT MASK

To address the non-differentiability of hard pruning, we adopt a *soft pruning* (Fang et al., 2024; Lin et al., 2024) approach, where each mask entry $m_j$ is relaxed to lie in $[0, 1]$ rather than being restricted to $\{0, 1\}$, i.e. $m_j = \sigma\left(\alpha(S_j - \text{Top}_k(S)\right) = \frac{1}{1+\exp(-\alpha(S_j - \text{Top}_k(S))}$, where $\sigma$ is the sigmoid function and $\alpha$ is a temperature hyperparameter that controls the steepness of $\sigma$. Under this relaxation, the gradient of the downstream loss $L_{\text{task}}$ w.r.t. the calibration weights $\mathbf{u}$ becomes

$$\frac{\partial L_{\text{task}}}{\partial u_i} = \sum_{j=1}^{d} \frac{\partial L_{\text{task}}}{\partial w'_j} \underbrace{\frac{\partial w'_j}{\partial m_j}}_{=w_j} \underbrace{\frac{\partial m_j}{\partial S_j}}_{\neq 0 \text{ (soft)}} \underbrace{\frac{\partial S_j}{\partial u_i}}_{\neq 0 \text{ in general}},$$

which is nonzero and therefore permits gradient-based optimization of $\mathbf{u}$.

**Stabilization via Perturbation.** In practice, we observed that direct optimization of $\mathbf{u}$ using the task loss can be highly sensitive (Bishop, 1995; Park et al., 2022), often resulting in poor sample selection

---

[1]For example, in Wanda $S(w_j) = |w_j| \|X\|_2$, where $\|X\|$ denotes the activation.

(also refer to ablation study in Section 5.4). To mitigate this instability, we introduce a perturbation at the input embedding level (Jiang et al., 2020; Zhu et al., 2020).

Let $\phi(x)$ denote the embedding of an input $x$, and let $\epsilon \in \mathbb{R}^{\dim(\phi(x))}$ be a random perturbation. We construct perturbed embeddings $\tilde{\phi}(x) = \phi(x) + \epsilon$, where $\epsilon$ is drawn from a uniform distribution following previous work (Aghajanyan et al., 2021). Abusing the notation $f$, the perturbed embeddings $\tilde{\phi}(x)$ are then passed through the model to compute our final task loss $L$:

$$L = L_{\text{pert}}(\mathbf{w}', \mathbf{u}) = \sum_{(x,y) \in \mathcal{D}_{\text{task}}} \ell\big(f(\tilde{\phi}(x); \mathbf{w}'), y\big),$$

whose gradient with respect to $u$ is used for updating the calibration weights. This perturbation acts as a regularizer, reducing sensitivity to local variations and promoting more robust calibration weight optimization. During training, $L$ is estimated on minibatches drawn from the calibration dataset, consistent with the zero-shot evaluation setting. A schematic overview is provided in Figure 4.

## 5 EXPERIMENT

We begin by outlining the experimental setup, followed by evaluation results on both structured pruning method and unstructured pruning method, and conclude with an ablation study. Additional experiments on Llama3.2-3B and Llama3.2-1B are provided in Appendix A.

### 5.1 EXPERIMENTAL SETUP

**Models.** We conduct our experiments on the Llama3 (Dubey et al., 2024) family of models (specifically on Llama3.1-8B) and on Qwen2.5-7B (et al., 2025) to demonstrate the effectiveness of our method across different model scales. All models are evaluated in a post-training, zero-shot setting to isolate the impact of the pruning process itself. Additional experiment results on Llama3.2-3B and Llama3.2-1B can be found in the Appendix.

**Pruning Baselines.** To assess the versatility and general applicability of OASIS, we apply our data selection method to three prominent pruning algorithms that represent different approaches to model compression: **Structured Pruning:** We use LLM-Pruner (Ma et al., 2023) and NIRVANA (Ai et al., 2025), two state-of-the-art methods that remove entire structural units (e.g., attention heads, FFN neurons). These methods are chosen for their practical relevance, as they can lead to direct computational speedups. **Unstructured Pruning:** We use Wanda (Sun et al., 2023), a widely-recognized magnitude-and-activation-based method that removes individual weights. This allows us to test whether OASIS can also benefit fine-grained pruning techniques. We set a target sparsity of 50% for all methods.

**Calibration Data Selection Baselines.** We compare OASIS against two competitive data selection baselines, in addition to the standard random sampling approach implicitly used by the original pruning methods: **Entropy-based Selection** (Kremer et al., 2014): A common heuristic that selects data samples with the highest token-level entropy, under the assumption that more complex and uncertain samples are more informative for calibration. **Synthetic Data Generation** (Ji et al., 2024): A recent approach where the LLM itself is prompted to generate its own calibration data. This strategy aims to create samples that are well-aligned with the model's internal knowledge distribution.

**Evaluation tasks and datasets.** We conduct experiments on two types of tasks. First, we follow (Ma et al., 2023) to evaluate zero-shot perplexity on WikiText2 (Merity et al., 2016), PTB (Wagner et al., 2020), and Lambada (Paperno et al., 2016). Second, we evaluate zero-shot accuracy on a suite of commonsense reasoning benchmarks, including ARC-easy (Clark et al., 2018), Winogrande (Sakaguchi et al., 2021), HellaSwag (Zellers et al., 2019), BoolQ (Clark et al., 2019) and PIQA (Bisk et al., 2020). All evaluation use the `lm-eval-harness` (Gao et al., 2024) framework.

**Implementation Details.** All our experiments were conducted on NVIDIA-GH200-120GB GPUs. Each selection method, including our proposed OASIS, selects a final subset of data from the same source to guide the pruning process, ensuring a fair comparison in terms of the amount of data used.

Table 1: Zero-shot performance of Llama3.1-8B and Qwen2.5-7B after applying 50% structured pruning. We compare baseline pruning methods against different calibration data selection strategies: entropy-based, synthetic data, and our proposed OASIS. **Bold** and underline denote the best and second-best results per group, respectively. ↓: lower is better. Δ: Average performance improvement.

| Method | WikiT↓ | PTB↓ | LambD↓ | ARC-e | WinoG | HellaS | BoolQ | PIQA | Δ |
|---|---|---|---|---|---|---|---|---|---|
| Llama-3.1-8B | 6.37 | 10.58 | 17.73 | 81.27 | 73.48 | 78.85 | 81.96 | 81.23 | |
| Nirvana | 39.94 | 59.96 | 58.12 | **40.27** | 56.04 | 42.14 | 60.76 | 63.28 | |
| + Entropy | 38.72 | 63.83 | 65.86 | 39.18 | 55.96 | **42.26** | 62.05 | 62.68 | |
| + Synthetic | 58.12 | 84.56 | 92.87 | 37.75 | 55.88 | 39.81 | 62.11 | 61.64 | |
| + OASIS (ours) | **36.37** | **52.92** | **52.92** | 39.18 | **58.01** | 42.14 | **62.17** | **63.33** | +0.47 |
| LLM-Pruner | 179.02 | 367.33 | **209.30** | 30.64 | **50.59** | 28.60 | 38.53 | 52.23 | |
| + Entropy | 179.02 | **295.15** | 277.27 | 30.26 | 49.33 | 27.93 | 37.95 | 52.99 | |
| + Synthetic | 416.23 | 624.84 | 403.43 | **31.78** | **50.59** | 27.99 | 37.83 | 52.77 | |
| + OASIS (ours) | **173.51** | 356.02 | 209.29 | 31.27 | 49.57 | **28.73** | **39.42** | **53.05** | +0.22 |
| Qwen2.5-7B | 6.89 | 12.18 | 20.09 | 77.36 | 73.16 | 78.96 | 84.65 | 78.84 | |
| Nirvana | 34.17 | 90.02 | **50.50** | 42.68 | 56.75 | 43.19 | 56.85 | 62.30 | |
| + Entropy | 35.81 | 84.56 | 77.00 | 39.02 | 53.12 | 37.65 | 61.59 | 60.17 | |
| + Synthetic | 74.63 | 143.85 | 112.03 | 38.80 | 51.14 | 36.68 | **61.68** | 57.45 | |
| + OASIS (ours) | **33.11** | **84.56** | 51.29 | **43.27** | **57.30** | **43.90** | 58.41 | **62.89** | +0.94 |
| LLM-Pruner† | **25.39** | **63.83** | **36.94** | **54.80** | 58.80 | 49.91 | 55.05 | **67.68** | |
| + Entropy | 90.01 | 190.57 | 163.00 | 42.21 | 48.30 | 36.13 | 55.69 | 60.23 | |
| + Synthetic | 179.02 | 391.02 | 295.15 | 43.22 | 50.51 | 37.07 | 42.23 | 59.41 | |
| + OASIS (ours) | **25.39** | **63.83** | **36.94** | 52.82 | **59.51** | **49.95** | **59.66** | 67.52 | +0.64 |

† Set sparsity to 40% due to observed degraded performance at 50%.

## 5.2 Results on Structured Pruning methods

The results for structured pruning, detailed in Table 1, highlight the significant impact of calibration data selection and the consistent advantages of our proposed method, OASIS. Across all models and pruning baselines, OASIS consistently outperforms or matches the state-of-the-art performance. For instance, when applied to NIRVANA on the Llama3.1-8B, OASIS achieves the lowest (best) perplexity on all three language modeling benchmarks. This superiority also translates to downstream tasks, where OASIS helps secure the best or second-best performance on nearly all common sense reasoning datasets. This trend is further confirmed on the Qwen2.5-7B model, where OASIS again delivers the top results for NIRVANA on both perplexity and the majority of downstream evaluations. This demonstrates OASIS's ability to select the optimized calibration data for different structured pruing methods and on different model backbones.

In contrast, the baseline selection methods show weaker and often unstable performance. The synthetic data approach is particularly unreliable for structured pruning. As seen with both NIRVANA and LLM-Pruner, this method frequently leads to a catastrophic increase in perplexity. This may be because the data generated by the model, while appearing coherent and assigned high confidence (low perplexity), might lack the structural diversity needed for coarse-grained pruning decisions. Such data could mislead the pruner into removing essential components. OASIS avoids this pitfall by selecting data based on its actual impact during the training process, thereby identifying a truly optimal and robust calibration set.

## 5.3 Results on Unstructured Pruning methods

As shown in Table 2, while unstructured pruning methods like Wanda are inherently more stable due to their fine-grained nature, a principled data selection strategy still offers clear benefits. OASIS consistently provides the best overall performance, demonstrating its effectiveness even in this less sensitive setting. For both the Llama3.1-8B and Qwen2.5-7B, OASIS achieves the best results on the majority of evaluation metrics. It secures the lowest perplexity on key benchmarks and pushes the state-of-the-art on downstream tasks, achieving the highest average improvement (Δ) among all selection methods.

Table 2: Zero-shot performance of Llama3.1-8B and Qwen2.5-7B after applying 50% unstructured pruning. We compare baseline pruning methods against different calibration data selection strategies: entropy-based, synthetic data, and our proposed OASIS. **Bold** and underline denote the best and second-best results per group, respectively. ↓: lower is better. Δ: Average performance improvement.

| Method | WikiT ↓ | PTB ↓ | LambD ↓ | ARC-e | WinoG | HellaS | BoolQ | PIQA | Δ |
|---|---|---|---|---|---|---|---|---|---|
| Llama-3.1-8B | 6.37 | 10.58 | 17.73 | 81.27 | 73.48 | 78.85 | 81.96 | 81.23 | |
| Wanda | 9.34 | 14.47 | 23.85 | 69.11 | 69.06 | 68.60 | 78.41 | 76.01 | |
| + Entropy | **9.20** | 14.47 | 24.23 | 69.02 | 70.40 | 68.72 | 78.35 | **76.55** | |
| + Synthetic | 9.34 | 14.47 | 25.00 | 68.18 | 69.53 | 68.58 | 77.65 | 75.73 | |
| + OASIS (ours) | 9.34 | 14.47 | **23.48** | **69.53** | **70.48** | **68.99** | **78.87** | 76.44 | +0.25 |
| Qwen2.5-7B | 6.89 | 12.18 | 20.09 | 77.36 | 73.16 | 78.96 | 84.65 | 78.84 | |
| Wanda | 8.52 | 15.51 | 23.51 | 73.44 | 69.77 | 70.92 | 82.35 | 77.69 | |
| + Entropy | 8.53 | 15.61 | 23.55 | 74.41 | 69.77 | 71.12 | 81.25 | 77.48 | |
| + Synthetic | 8.77 | 15.74 | 24.38 | 73.78 | 69.85 | 70.44 | **83.88** | 76.88 | |
| + OASIS (ours) | **8.50** | **15.40** | **23.48** | **74.66** | **70.56** | **71.32** | 82.75 | **78.02** | +0.49 |

The baseline methods, while not failing as catastrophically as in the structured pruning case, are generally weaker. The entropy method, for instance, often results in slightly higher perplexity and middling downstream performance. This suggests that even for weight-level decisions, data that seems "hard" for the model may not be the most informative for identifying parameter importance. By optimizing the selection process directly, OASIS identifies a calibration set that provides a more effective signal, leading to modest but consistent performance gains that heuristic and generative methods cannot reliably achieve.

### 5.4 ABLATION STUDY

To better understand the effect of perturbation, we conduct an ablation study on a controlled group of golden, mediocre, and detrimental samples. The goal is to verify whether the target loss can discriminate high-quality data under different conditions. Figure 5 presents the evolution of data weights over training steps. Without noise, the optimization is unstable: the golden samples are quickly overshadowed, mediocre ones dominate, and even the detrimental samples can receive larger weights than the golden ones. This indicates that the raw loss alone is insufficiently discriminative. By contrast, when uniform noise is injected at the embedding level, the weighting process becomes more robust: golden samples are consistently amplified, while mediocre and detrimental samples are gradually suppressed. This demonstrates that embedding perturbation is not a cosmetic trick but an essential component for stabilizing training and enabling reliable data selection in downstream tasks.

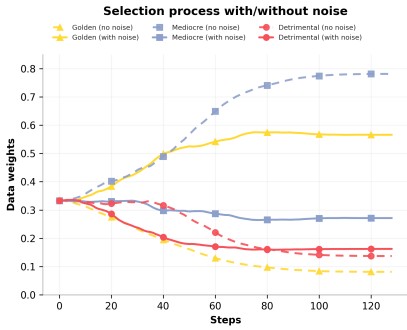

Figure 5: Ablation Study on Noise Effect

### 6 CONCLUSION

In this work, we addressed the critical yet inconsistent nature of calibration data selection for LLM pruning. We extended prior analyses to demonstrate not only that data quality is hierarchical, but also that a sample's quality is highly context-dependent on the specific model and pruning method. This discovery reveals that simple heuristic or filtering approaches are fundamentally insufficient.

To solve this, we introduced OASIS, which represents a paradigm shift from heuristic pre-processing to an end-to-end, trainable optimization problem. By leveraging a differentiable soft-mask proxy, OASIS creates a novel pipeline that directly optimizes the data subset against the final pruned model's performance. This transforms data selection from a heuristic gamble into a principled process. Experiments confirm that OASIS enhances diverse state-of-the-art pruning methods, establishing a more effective and reliable standard for data-aware model compression.

## ETHICS STATEMENT

Our work focuses on the algorithmic improvement of model pruning, a technique aimed at increasing the computational efficiency of Large Language Models (LLMs). The primary ethical benefit of this research is positive: by making models smaller and faster, our method contributes to reducing the energy consumption, computational cost, and hardware requirements for deploying and running LLMs. This helps to democratize access to powerful AI technologies and lowers their environmental impact.

We exclusively use publicly available, pre-trained models and standard, open-source datasets. As such, our work does not involve human subjects, private data, or the creation of new datasets. We acknowledge that the LLMs used in our experiments inherit the potential biases, limitations, and societal risks of their original training data. Our method does not aim to mitigate these underlying issues but rather to compress the models as they are. The selection algorithm itself is task-agnostic and does not inherently introduce new biases beyond those potentially amplified by the pruning process itself, an area that warrants further study across all compression techniques. We have adhered to the ICLR Code of Ethics throughout this research.

## REPRODUCIBILITY STATEMENT

We are committed to ensuring the reproducibility of our research. To facilitate this, we have uploaded the source code to an anonymous link https://anonymous.4open.science/r/OASIS-DC05/. The core algorithm of our proposed method is detailed in Section 4, with specific implementation details, hyperparameters, and the computational environment described in Section 5.1. All baseline models and pruning algorithms are publicly available and properly cited. The datasets used for calibration and evaluation are standard benchmarks in the field. Detailed experimental results are presented in Section 5.

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

Table 3: Zero-shot performance of Llama3.2-3B after applying 50% structured pruning. We compare baseline pruning methods against versions enhanced with different calibration data selection strategies: entropy-based, synthetic data, and our proposed OASIS. **Bold** and underline denote the best and second-best results per group, respectively. ↓: lower is better.

| Method | WikiT ↓ | PTB ↓ | LambD ↓ | ARC-e | WinoG | HellaS | BoolQ | PIQA | Δ |
|---|---|---|---|---|---|---|---|---|---|
| Llama-3.2-3B | 7.87 | 12.57 | 20.09 | 71.68 | 69.06 | 73.69 | 72.78 | 77.48 | |
| Nirvana | 61.87 | 112.03 | 105.24 | 33.80 | 51.78 | 34.75 | 57.28 | **59.03** | |
| + Entropy | 61.87 | 115.58 | 105.24 | 32.45 | 51.38 | 34.27 | 47.71 | 57.94 | |
| + Synthetic | 95.82 | 168.17 | 168.17 | 32.20 | **54.30** | 34.04 | 50.64 | 57.78 | |
| + OASIS (ours) | **58.12** | **108.58** | **102.00** | **34.13** | 51.54 | **35.12** | 59.11 | 58.75 | +0.41 |
| LLM-Pruner | **196.62** | 268.74 | 179.02 | 31.73 | 46.72 | **28.92** | 60.37 | 56.47 | |
| + Entropy | 334.45 | 356.02 | 260.47 | **32.66** | 48.30 | 28.20 | 52.11 | 54.52 | |
| + Synthetic | 938.00 | 605.62 | 486.63 | 30.89 | 48.54 | 28.09 | 48.72 | 54.30 | |
| + OASIS (ours) | **196.62** | **244.69** | **168.17** | 31.81 | **48.70** | 28.64 | 59.14 | 57.02 | +0.22 |

Table 4: Zero-shot performance of Llama-3.2-1B after applying 50% unstructured pruning. We compare baseline pruning methods against versions enhanced with different calibration data selection strategies: entropy-based, synthetic data, and our proposed OASIS. **Bold** denotes the best results per group. ↓: lower is better.

| Method | WikiT ↓ | PTB ↓ | LambD ↓ | ARC-e | WinoG | HellaS | BoolQ | PIQA | Δ |
|---|---|---|---|---|---|---|---|---|---|
| Llama-3.2-1B | 9.64 | 16.65 | 23.48 | 60.27 | 59.98 | 63.66 | 63.88 | 74.27 | |
| Wanda | 21.05 | 38.72 | 50.50 | 49.92 | 55.49 | 45.01 | 62.02 | 66.10 | |
| + Entropy | 21.05 | 38.72 | 52.10 | 50.21 | 55.72 | **45.24** | 61.37 | 66.10 | |
| + Synthetic | 21.72 | 39.33 | 53.75 | 50.67 | 54.93 | 45.15 | 61.90 | 66.38 | |
| + OASIS(Ours) | 21.05 | **38.11** | 50.50 | **50.72** | **55.96** | 44.98 | **62.48** | **66.65** | +0.35 |

# A    ADDITIONAL RESULTS ON LLAMA3.2-3B AND LLAMA3.2-1B

In the main content, we provide the experiment results on Llama3.1-8B and Qwen2.5-7B. Here, we present additional results on Llama3.2-3B and Llama3.2-1B. The results presented in Table 3 and Table 4 on the Llama-3.2 3B and 1B models further validate the conclusions drawn in the main paper. These experiments demonstrate that the effectiveness of OASIS is consistent across different model sizes within the Llama family.

As shown in Table 3, the structured pruning results on the 3B model mirror the trends observed on the larger 8B model. (1) Consistent Superiority of OASIS: For both NIRVANA and LLM-Pruner, OASIS delivers the most significant improvements in language modeling performance, achieving the lowest (best) perplexity scores on nearly all benchmarks. This is a critical result, as perplexity is a strong indicator of a model's fundamental language understanding, which is often severely damaged by structured pruning. OASIS proves most effective at preserving this core capability. (2) Instability of Baseline Selection Methods: The limitations of heuristic and generative methods are again evident. The synthetic data approach, in particular, leads to a catastrophic degradation for both NIRVANA and LLM-Pruner, with perplexity scores increasing by a factor of 2-5x compared to the baseline. This reinforces our hypothesis that data which is merely "low perplexity" for the model is not necessarily informative for making coarse-grained pruning decisions and can be severely misleading. (3) Downstream Task Performance: While the baseline Nirvana model without any specialized selection performs surprisingly well on some downstream tasks, OASIS provides the most robust and balanced performance profile. It secures the top results on several tasks (ARC-e, HellaS, BoolQ) while remaining competitive on others, resulting in the highest average performance boost (Δ).

The results for Wanda on the Llama-3.2-1B model, presented in Table 4, confirm that OASIS remains beneficial even for less sensitive, fine-grained pruning methods. (1) Modest but Consistent Gains: While the performance differences are much smaller compared to structured pruning, OASIS consistently outperforms the other selection strategies on the majority of downstream tasks. It achieves the highest scores on ARC-e, WinoG, and BoolQ, leading to the largest overall performance boost (Δ=+0.35). (2) Minimal Impact on Perplexity: For unstructured pruning at 50% sparsity,

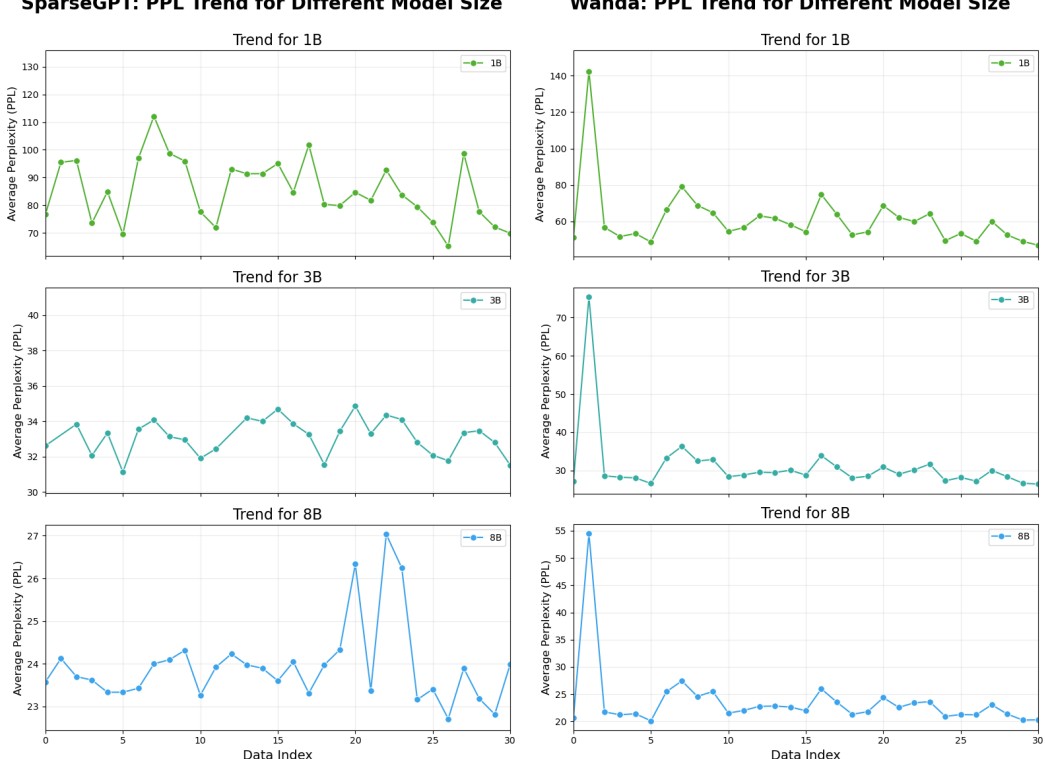

Figure 6: Perplexity (PPL) trends after pruning with individual calibration samples. (Left Column) SparseGPT results on Llama 1B, 3B, and 8B models. (Right Column) Wanda results on the same models. The x-axis indexes the different data samples used for calibration. The plots reveal that data preference is highly dependent on both the pruning method and, in some cases, the model size.

the impact on perplexity is generally low across all methods. However, OASIS still manages to achieve a slight edge, securing the best performance on PTB. This suggests that even when the overall degradation is minimal, a principled selection method can still find a better optimization path than heuristic approaches. (3) Overall, these additional experiments strengthen our central claim that OASIS is a versatile and robust tool for enhancing post-training pruning. It provides substantial benefits where they are needed most (structured pruning) while still offering consistent, marginal gains in less sensitive scenarios (unstructured pruning).

## B  Context-Dependency of Calibration Data Preference

In the main paper, we argue that the quality of a calibration sample is not a universal property but is context-dependent on the pruning method and model. To provide evidence for this claim, we conduct an additional analysis on two widely-used unstructured pruning methods, Wanda and SparseGPT (this is to minimize the variance brought in by structured pruning VS unstructured pruning), across the Llama model family (1B, 3B, and 8B).

The results in Figure 6 highlight two key aspects of this context-dependency: (1) **Dependence on Pruning Method:** The "best" and "worst" calibration samples are clearly different for Wanda and SparseGPT. For example, sample #1 is consistently the most detrimental for Wanda across all model sizes, causing a massive spike in perplexity. However, for SparseGPT, sample #1 performs reasonably well. (2) **Dependence on Model Architecture/Size:** The consistency of data preference across a model family also varies by method. For Wanda (right column), the performance trend is remarkably consistent across the 1B, 3B, and 8B models; a sample that is good for the 1B model is generally also good for the 8B model. In contrast, for SparseGPT (left column), the trend is much less consistent. For instance, sample #7 yields one of the worst PPLs for the 1B model but delivers

Table 5: Data samples of NIRVANA

| Pruning Method | Data Example |
|---|---|
| NIRVANA | **Golden:** was caroling out to me : baby, baby, my sweet angel baby, baby, my perfect angel baby, baby, my amazing angel baby, babythe one who has stolen and seized my heart there is something special about your lovely eyes something special about the curving and shapeliness of your lips something golden about the way you smile something brilliant about the way you talk baby, baby, i am falling so madly for you baby, baby, i cant go on without thinking about you baby, baby, you are my heart and my breath itself baby, babyyou are the one i live to see and eternally hold |
| | **Mediocre:** find in these entries : more on the flora and fauna of deucado and helome a discourse on the nature and cure of type 1 diabetes a complete study on methods of exploiting solar energy available today more backstories the nature of heaven and the hyper-verse the original short story last of the cavaliers which became the historical section folded into the ark lords hot fusion versus cold fusion how i was foiled by the simpsons movie and stephen kings under the dome the untold story of the arrival of the deucadons at tau ceti an actual picture of minimcom before he became a starship the design of the split |
| | **Detrimental:** step ahead by phoebe laplume a dark shadow would cross the bottom of the staircase to and from the living room and the kitchen horror 61st of september part one by gary murphy the great terror event levelled most of new york city in 2025 science fiction the litter by daniel davis the worst times in the house were when barry and i were there alone horror androids attack by kevin l jones he had been dispatched to liquidate the rebels with an experimental army of android soldiers science fiction the days of mr thomas by james rhodes returns next week dystopia the book of the thousand |

average performance on the 3B and 8B models. We hypothesize this difference in stability stems from the nature of their saliency scores. Wanda uses a zero-order metric, $S = \|\mathbf{W}\mathbf{X}^2\|$, which is based on parameter weights and input activations, making it relatively stable. In contrast, SparseGPT's score, $S = \|\mathbf{W}\|^2/\text{diag}(\mathbf{H})$, is a second-order metric that incorporates the Hessian matrix ($\mathbf{H}$). This inclusion of higher-order information, which captures the curvature of the loss function, makes its importance estimation more complex and sensitive to the specific model state and calibration data.

Together, these results strongly support our central claim: the notion of a "golden" calibration sample is not static. It is a function of the complex interaction between the data, the pruning algorithm, and the model architecture.

## C  QUALITATIVE EXAMPLES OF SELECTED DATA TIERS

To provide a more qualitative understanding of the data selected by OASIS, this section presents examples from the Golden, Mediocre, and Detrimental tiers for both Wanda and NIRVANA. These examples highlight how the definition of a "good" calibration sample is highly dependent on the pruning method's underlying saliency metric.

For a structured pruning method like NIRVANA, the qualitative characteristics of each tier are less intuitive from a human perspective: The **Golden** sample is a repetitive poem. This seems counter-intuitive. However, for a structured method that aggregates importance across entire components (like attention heads or FFN layers), this type of patterned, sentimental text might serve as an effective "stress test." It may force specific structural blocks responsible for pattern recognition and sentiment processing to work in a highly coordinated manner, providing a strong, clear signal about the importance of these entire blocks. The **Detrimental** sample, a list of varied book descriptions, appears more diverse than the golden sample. However, it's possible these short, disconnected

snippets activate a wide range of components but only superficially. This shallow activation across many structures may not provide a strong enough signal for the aggregator to determine which entire components are critical, leading to suboptimal pruning decisions.

As for Wanda, the qualitative differences are intuitive and easy to interpret: The **Golden** sample is a linguistically complex and diverse paragraph. This type of text likely activates a broad and representative set of pathways throughout the model, providing a well-rounded signal for which weights are generally important. The **Mediocre** sample, a long list of items, falls in between. It has more vocabulary diversity than the detrimental sample but lacks the syntactic complexity of the golden one, providing a less comprehensive signal. The **Detrimental** sample is extremely repetitive. This text would cause a very small and specific set of neurons and weights to activate with high magnitude repeatedly. Using this as a calibration signal would give a highly biased view, leading the pruner to mistakenly preserve weights associated with the repeated phrase while removing others that are more generally useful.

Table 6: Data samples of Wanda

| Pruning Method | Data Example |
|---|---|
| Wanda | **Golden:** the limitations observed in current scientific understanding, no vehicle would ever be able to travel faster than ten kilometres per haca sure as no tree will ever be able to grow more than ten items of fruit per year, eleven being an unlucky number and thus perpetually avoided in nature ; sure as protein can only ever come from the remains of slaughtered animals ; and sure as there will never be any evidence in favour of life outside glix before pushing his foot down upon the spike the pedal having already been invented, but naturally rejected in favour of the electroconductive abilities of the spike and hurtling forward into a brick wall at 57.11km |
| | **Mediocre:** , military clothing, load bearing belts and harnesses, military backpacks, 30 pairs of civilian casual pants, shorts, sleeveless shirts, 15 pairs of athletic shoes, 100 pairs of athletic socks, 8 athletic supports, 24 pairs of thermal underwear, 24 pairs of thermal socks, 6 pairs of cold weather boots, military-style hot weather clothing, 60 pairs of gloves military work-style, 10 containers of military-style sanitary gloves, 6 pairs of cold weather gloves, 10 laundry bags, disposal surgical gloves, military-style warm weather jackets, military-style cold weather jackets, civilian-style warm and cold |
| | **Detrimental:** , i love you, i love you, i love you, i love you, i love you, i love you, i love you, i love you, i love you, i love you, i love you, i love you, i love you, i love you, i love you, i love you, i love you, i love you, i love you, i love you, i love you, i love you, i love you, i love you, i love you, i love you, i love you, i love you, i love you, i love you, i love you, i love you, i love you, i love you, i love you |

# D  DETAILS ON MICRO-LEVEL ANALYSIS

In Section 3.2, we provide the micro analysis on calibration data. Here, we provide the details of that analysis in the following tables, with an additional result on SparseGPT.

Table 7: Synergistic and Degradative Effects of Combining Calibration Samples from Different Quality Tiers for the **Wanda** Pruning Method. The change ($\Delta$) is calculated as (Combination PPL - Best Individual PPL). Negative values (green) indicate synergistic improvement, while positive values (red) indicate performance degradation.

| Category | Sample 1 (PPL) | Sample 2 (PPL) | Combination PPL | Change ($\Delta$) | Observed Effect |
|---|---|---|---|---|---|
| **Gold + Gold** | 77 (45.8) | 111 (45.9) | **45.2** | -0.6 | Synergy |
| | 91 (46.8) | 103 (46.6) | **43.4** | -3.2 | Strong Synergy |
| | 77 (45.8) | 103 (46.6) | **44.2** | -1.6 | Synergy |
| | 77 (45.8) | 91 (46.8) | **44.5** | -1.3 | Synergy |
| | 91 (46.8) | 110 (46.8) | **44.8** | -2.0 | Strong Synergy |
| **Mediocre + Mediocre** | 9 (64.8) | 23 (64.4) | **57.7** | -6.7 | Strong Synergy |
| | 23 (64.4) | 54 (67.1) | **59.0** | -5.4 | Strong Synergy |
| | 23 (64.4) | 33 (64.5) | **57.4** | -7.0 | Strong Synergy |
| | 8 (68.8) | 16 (74.8) | **64.5** | -4.3 | Synergy |
| | 20 (68.7) | 54 (67.1) | **60.5** | -6.6 | Strong Synergy |
| **Bad + Bad** | 1 (142.3) | 66 (148.6) | **111.0** | -31.3 | Improvement |
| | 1 (142.3) | 95 (117.3) | **95.9** | -21.4 | Improvement |
| | 1 (142.3) | 86 (115.4) | **93.4** | -22.0 | Improvement |
| | 66 (148.6) | 95 (117.3) | **116.5** | -0.8 | Slight Improvement |
| | 66 (148.6) | 86 (115.4) | 115.8 | +0.4 | Neutral / No Effect |
| **Gold + Mediocre** | 77 (45.8) | 58 (70.5) | 49.5 | +3.7 | Degradation |
| | 111 (45.9) | 17 (63.9) | 48.6 | +2.7 | Degradation |
| | 115 (46.5) | 33 (64.5) | 48.3 | +1.8 | Degradation |
| | 103 (46.6) | 61 (64.8) | 47.2 | +0.6 | Slight Degradation |
| | 110 (46.8) | 6 (66.4) | **45.9** | -0.9 | Synergy (Outlier) |
| **Gold + Bad** | 77 (45.8) | 66 (148.6) | 53.5 | +7.7 | Strong Degradation |
| | 111 (45.9) | 66 (148.6) | 53.5 | +7.6 | Strong Degradation |
| | 115 (46.5) | 66 (148.6) | 53.2 | +6.7 | Strong Degradation |
| | 103 (46.6) | 1 (142.3) | 53.5 | +6.9 | Strong Degradation |
| | 110 (46.8) | 1 (142.3) | 53.0 | +6.2 | Strong Degradation |
| **Mediocre + Bad** | 9 (64.8) | 1 (142.3) | 70.5 | +5.7 | Degradation |
| | 23 (64.4) | 1 (142.3) | 70.5 | +6.1 | Degradation |
| | 8 (68.8) | 66 (148.6) | 76.1 | +7.3 | Strong Degradation |
| | 33 (64.5) | 66 (148.6) | 71.7 | +7.2 | Strong Degradation |
| | 20 (68.7) | 95 (117.3) | 73.5 | +4.8 | Degradation |

Table 8: Synergistic Effects of Combining Multiple Golden Samples. The "Best Individual PPL" column serves as the baseline for each group. The Synergy ($\Delta$) column, calculated as (Combination PPL - Best Individual PPL), quantifies the performance gain from the combination. All $\Delta$ values are negative, indicating a consistent synergistic effect. Random 10 samples avgppl is 43.3.

| Category | # Samples ($k$) | Sample IDs in Combination | Best Individual PPL | Combination PPL | Synergy ($\Delta$) |
|---|---|---|---|---|---|
| **Golden Trio** ($k=3$) | 3 | 77, 103, 111 | 45.8 | **44.1** | -1.7 |
| | 3 | 77, 91, 111 | 45.8 | **44.4** | -1.4 |
| | 3 | 77, 91, 103 | 45.8 | **43.4** | -2.4 |
| | 3 | 91, 103, 111 | 45.9 | **43.9** | -2.0 |
| | 3 | 77, 93, 110 | 45.8 | **43.1** | -2.7 |
| **Golden Quad** ($k=4$) | 4 | 77, 91, 103, 111 | 45.8 | **43.0** | -2.8 |
| | 4 | 77, 93, 103, 110 | 45.8 | **43.4** | -2.4 |
| | 4 | 77, 93, 107, 111 | 45.8 | **42.3** | -3.5 |
| | 4 | 91, 93, 107, 115 | 46.5 | **44.2** | -2.3 |
| | 4 | 91, 103, 111, 115 | 45.9 | **44.2** | -1.7 |
| **Golden All-In** | 10 | 30, 77, 91, 93, 101, 103, 107, 110, 111, 115 | 45.8 | **42.6** | -3.2 |

Table 9: Synergistic and Degradative Effects on the **SparseGPT** Method. Note the inconsistent effect in the "Gold + Gold" category and the synergistic effect in some "Mediocre + Bad" cases, which differ from the Wanda method's behavior.

| Category | Sample 1 (PPL) | Sample 2 (PPL) | Combination PPL | Change (Δ) | Observed Effect |
|---|---|---|---|---|---|
| **Gold + Gold** | 26 (65.2) | 100 (65.4) | **62.8** | -2.4 | Synergy |
| | 26 (65.2) | 94 (64.4) | **63.1** | -1.3 | Synergy |
| | 100 (65.4) | 106 (65.1) | 67.5 | +2.4 | Degradation |
| | 26 (65.2) | 106 (65.1) | 66.6 | +1.5 | Degradation |
| | 94 (64.4) | 106 (65.1) | **63.8** | -0.6 | Synergy |
| **Mediocre + Mediocre** | 51 (84.3) | 56 (84.1) | **79.9** | -4.2 | Synergy |
| | 18 (80.3) | 97 (80.0) | **68.3** | -11.7 | Strong Synergy |
| | 18 (80.3) | 31 (80.3) | **76.8** | -3.5 | Synergy |
| | 21 (81.7) | 58 (81.8) | **73.9** | -7.8 | Strong Synergy |
| | 23 (83.7) | 83 (82.5) | **80.6** | -1.9 | Synergy |
| **Bad + Bad** | 84 (120.6) | 85 (123.2) | **114.6** | -6.0 | Improvement |
| | 67 (132.5) | 84 (120.6) | **116.9** | -3.7 | Improvement |
| | 67 (132.5) | 85 (123.2) | **109.8** | -13.4 | Strong Improvement |
| | 67 (132.5) | 95 (119.3) | **106.1** | -13.2 | Strong Improvement |
| | 67 (132.5) | 86 (119.2) | **106.7** | -12.5 | Strong Improvement |
| **Gold + Mediocre** | 26 (65.2) | 58 (81.8) | 70.2 | +5.0 | Degradation |
| | 26 (65.2) | 81 (84.0) | 75.3 | +10.1 | Strong Degradation |
| | 94 (64.4) | 4 (84.9) | 65.3 | +0.9 | Slight Degradation |
| | 94 (64.4) | 49 (85.4) | 68.7 | +4.3 | Degradation |
| | 100 (65.4) | 103 (82.4) | **65.3** | -0.1 | Neutral / No Effect |
| **Gold + Bad** | 26 (65.2) | 67 (132.5) | 68.6 | +3.4 | Degradation |
| | 94 (64.4) | 67 (132.5) | 67.7 | +3.3 | Degradation |
| | 100 (65.4) | 85 (123.2) | 82.0 | +16.6 | Strong Degradation |
| | 94 (64.4) | 85 (123.2) | 69.0 | +4.6 | Degradation |
| | 106 (65.1) | 84 (120.6) | 71.0 | +5.9 | Degradation |
| **Mediocre + Bad** | 31 (80.3) | 67 (132.5) | 81.3 | +1.0 | Slight Degradation |
| | 21 (81.7) | 67 (132.5) | **74.9** | -6.8 | Synergy |
| | 56 (84.1) | 85 (123.2) | **82.0** | -2.1 | Synergy |
| | 18 (80.3) | 84 (120.6) | **76.8** | -3.5 | Synergy |
| | 81 (84.0) | 85 (123.2) | **81.5** | -2.5 | Synergy |

Table 10: Synergistic Effects of Combining Multiple Golden Samples for the **SparseGPT** Method. Note the "less is more" effect, where the best 4-sample combination outperforms the 10-sample combination. Random 10 samples avgppl is 61.0.

| Category | # Samples (k) | Sample IDs in Combination | Best Individual PPL | Combination PPL | Synergy (Δ) |
|---|---|---|---|---|---|
| **Golden Trio (k = 3)** | 3 | 26, 94, 100 | 64.4 | **62.8** | -1.6 |
| | 3 | 26, 94, 106 | 64.4 | **62.7** | -1.7 |
| | 3 | 94, 100, 106 | 64.4 | **63.8** | -0.6 |
| | 3 | 26, 90, 106 | 65.1 | **61.6** | -3.5 |
| | 3 | 90, 94, 106 | 64.4 | 65.2 | +0.8 |
| **Golden Quad (k = 4)** | 4 | 26, 94, 100, 106 | 64.4 | **63.0** | -1.4 |
| | 4 | 94, 100, 105, 106 | 64.4 | **63.8** | -0.6 |
| | 4 | 26, 90, 94, 100 | 64.4 | **61.3** | -3.1 |
| | 4 | 90, 94, 100, 106 | 64.4 | **60.6** | -3.8 |
| | 4 | 94, 100, 105, 106 | 64.4 | **63.8** | -0.6 |
| **Golden All-In** | 10 | 5, 26, 90, 91, 94, 100, 105, 106, 113, 114 | 64.4 | **60.7** | -3.7 |

# E    THE USE OF LARGE LANGUAGE MODELS (LLMs)

We use LLMs in this work in order to revise the paper writing and check for grammar errors.

Table 11: Analysis of Data Combination Effects on NIRVANA. Note the inconsistent effects in "Gold + Gold" and the unstable behavior in "Bad + Bad" categories.

| Category | Sample 1 (PPL) | Sample 2 (PPL) | Combination PPL | Change (Δ) | Observed Effect |
|---|---|---|---|---|---|
| **Gold + Gold** | 111 (168.4) | 110 (169.3) | 175.6 | +7.2 | Degradation |
| | 111 (168.4) | 108 (173.7) | **166.6** | -1.8 | Synergy |
| | 110 (169.3) | 108 (173.7) | 170.9 | +1.6 | Degradation |
| | 111 (168.4) | 55 (174.4) | **165.0** | -3.4 | Strong Synergy |
| | 110 (169.3) | 55 (174.4) | 172.6 | +3.3 | Degradation |
| **Mediocre + Mediocre** | 8 (230.1) | 5 (232.0) | **218.5** | -11.6 | Strong Synergy |
| | 5 (232.0) | 3 (233.2) | **225.6** | -6.4 | Synergy |
| | 10 (232.3) | 114 (232.5) | **209.4** | -22.9 | Strong Synergy |
| | 15 (235.0) | 78 (237.1) | **205.3** | -29.7 | Strong Synergy |
| | 3 (233.2) | 61 (239.8) | **223.3** | -9.9 | Strong Synergy |
| **Bad + Bad** | 54 (473.6) | 62 (426.9) | 528.8 | +101.9 | Catastrophic Degradation |
| | 102 (423.4) | 54 (473.6) | 444.6 | +21.2 | Strong Degradation |
| | 62 (426.9) | 102 (423.4) | **392.7** | -30.7 | Strong Improvement |
| | 43 (418.4) | 54 (473.6) | 602.3 | +183.9 | Catastrophic Degradation |
| | 43 (418.4) | 62 (426.9) | **342.4** | -76.0 | Strong Improvement |
| **Gold + Mediocre** | 111 (168.4) | 8 (230.2) | 182.9 | +14.5 | Strong Degradation |
| | 111 (168.4) | 115 (230.7) | 183.2 | +14.8 | Strong Degradation |
| | 110 (169.3) | 78 (237.1) | 187.0 | +17.7 | Strong Degradation |
| | 110 (169.3) | 97 (246.3) | 193.2 | +23.9 | Strong Degradation |
| | 108 (173.7) | 8 (230.2) | 207.1 | +33.4 | Strong Degradation |
| **Gold + Bad** | 111 (168.4) | 54 (473.6) | 315.8 | +147.4 | Severe Degradation |
| | 111 (168.4) | 62 (426.9) | 280.6 | +112.2 | Severe Degradation |
| | 110 (169.3) | 54 (473.6) | 347.9 | +178.6 | Severe Degradation |
| | 110 (169.3) | 6 (325.2) | 304.8 | +135.5 | Severe Degradation |
| | 108 (173.7) | 102 (423.4) | 280.6 | +106.9 | Severe Degradation |
| **Mediocre + Bad** | 15 (235.0) | 54 (473.6) | 368.5 | +133.5 | Severe Degradation |
| | 114 (232.5) | 54 (473.6) | 349.7 | +117.2 | Severe Degradation |
| | 8 (230.2) | 54 (473.6) | 345.0 | +114.8 | Severe Degradation |
| | 3 (233.2) | 62 (426.9) | 331.2 | +98.0 | Severe Degradation |
| | 71 (232.6) | 62 (426.9) | 352.8 | +120.2 | Severe Degradation |

Table 12: Synergistic Effects of Combining Multiple Golden Samples for NIRVANA. The "less is more" effect is highly pronounced, with the 10-sample combination performing significantly worse than smaller, well-chosen sets.

| Category | # Samples (k) | Sample IDs in Combination | Best Individual PPL | Combination PPL | Synergy (Δ) |
|---|---|---|---|---|---|
| **Golden Trio (k = 3)** | 3 | 111, 110, 108 | 168.4 | 178.0 | +9.6 |
| | 3 | 110, 108, 55 | 169.3 | **168.9** | -0.4 |
| | 3 | 111, 110, 55 | 168.4 | **165.2** | -3.2 |
| | 3 | 111, 110, 59 | 168.4 | 189.1 | +20.7 |
| | 3 | 108, 55, 59 | 173.7 | 191.0 | +17.3 |
| **Golden Quad (k = 4)** | 4 | 111, 110, 108, 55 | 168.4 | **167.2** | -1.2 |
| | 4 | 111, 110, 108, 59 | 168.4 | 181.5 | +13.1 |
| | 4 | 111, 110, 55, 59 | 168.4 | 175.9 | +7.5 |
| | 4 | 110, 108, 55, 59 | 169.3 | 178.0 | +8.7 |
| | 4 | 110, 108, 55, 59 | 169.3 | 175.9 | +6.6 |
| **Golden All-In** | 10 | 111, 110, 108, 55, 59, 64, 31, 37, 56, 39 | 168.4 | 197.1 | +28.7 |

