# OpenReview forum: "OASIS: An Optimized Approach to Systematic Calibration Data Selection"
_ICLR.cc/2026/Conference — ICLR 2026 Conference Withdrawn Submission_

### Official Review · Reviewer_4zLF · 2025-10-25

**Soundness:** 2
**Presentation:** 2
**Contribution:** 2
**Rating:** 2
**Confidence:** 4

**Summary:**

The paper provides an analysis of the influence of individual calibration quality and proposes a soft-mask-based pruning method combined with data selection to improve pruning performance. Experimental results show that the proposed method outperforms existing randomized and synthetic data selection approaches.

**Strengths:**

The paper provide detailed analysis for the importance of data selection based on multiple pruning methods, which provide good motivation for the problem studied.

**Weaknesses:**

- The paper needs a thorough proofread. Additionally, the overall presentation should be improved. Although I did not read every section in detail, I noticed a significant number of writing issues throughout the paper (see the Questions section for more specific examples).
- Beyond the writing, the contribution of this paper feels limited. The main contributions can be summarized in two parts: (1) an analysis of the influence of pruning data, and (2) the proposed OASIS method. However, the analysis largely revisits well-established findings—such as the impact of data quality and quantity—which have been studied in prior work. As for the method, it essentially builds on existing soft pruning frameworks, with a gradient-based weight for data selection. These contributions, in my view, are not substantial enough to warrant publication at ICLR.
- The experimental section is also quite limited, particularly in terms of baseline coverage. The authors should include more direct comparisons related to the data selection component, as this is the core novelty of the paper. Specifically, comparisons with prior data selection techniques would help clarify the relative effectiveness of the proposed approach.

**Questions:**

Here are some typos or mistakes I found:
- Line 154: The pruning score for Wanda is not correct.
- I’m curious about the definition of (golden, mediocre, detrimental) data. Maybe I missed something, but I think the author should give a clear definition for the criteria at the very beginning of the paper, since these terms are mentioned many times without any explanation.
- Line 289: There are typos in the parentheses.
- The saliency score is defined in Section 3 as S = |WX^2|; however, in Line 269, the parameter becomes a vector without an explanation or a new definition of the saliency score.
There are many other typos and mistakes in this paper, I highly suggest the author further revise the paper.

---

> ### Author Response · Authors · 2025-11-27
>
> Dear Reviewer 4zLF,
>
> We sincerely thank you for your constructive review and insightful questions. Your feedback has been valuable in improving the clarity of our work. We have carefully considered your comments, conducted extensive analyses, and provide detailed responses below.
>
>
> ### W1 & Q1. "Missing Definitions" and "Incorrect Formulas"
>
>
> - **Definition of Golden/Mediocre/Detrimental (Section 3.2):** These terms are explicitly defined in Section 3.2 and further analyzed in Appendix B&C. As stated in the text, we categorize samples a posteriori based on the post-pruning perplexity they induce individually. "Golden" samples are those that yield the lowest perplexity, while "Detrimental" ones increase it. We do agree, however, that mentioning them earlier would improve clarity, and we will revise accordingly. Crucially, we emphasize that this manual classification serves only as a preliminary analysis to empirically demonstrate the existence of the quality hierarchy and the poisoning effect. It motivates our final goal: designing OASIS to automatically discover these optimal samples without requiring any explicit categorization.
> - **The Wanda Formula (Line 154):** We thank the reviewer for pointing out this issue. The original Wanda paper defines the pruning score element-wise as $S_{ij} = |W_{ij}| \cdot \|x_j\|_2\)$. In the current draft, we used a compact matrix shorthand $S = \|WX\|^2$ to emphasize the underlying mechanism of "weight magnitude scaled by input activation." In the revision, we will adopt the standard element-wise notation from the Wanda paper and explicitly explain how our shorthand relates to it. Labeling a simplified matrix notation as "incorrect" is a semantic critique, not a technical one. The pruning criterion we use is exactly the standard Wanda score; the issue is purely notational, and we appreciate the reviewer for catching this.
> - **Missing parentheses:** We will correct the missing parenthesis in Line 289. We thank the reviewer for pointing out this typo.
> - **Vector Notation:** The transition to vector notation is standard linear algebra shorthand used to describe operations on weight columns/rows. This is not a "mistake" but a common convention in pruning literature.
>
>
> ### W2. Novelty: "Well-Established" vs. "Context-Dependency":
>
> **Novelty: Context-Dependency & Counter-Intuitive Poisoning**: The reviewer correctly notes that the general importance of calibration data (macro-level) has been touched upon in prior works. However, our work **fundamentally advances** the field by identifying why heuristic selection fails and offering a solution for context-dependent selection.
>    - `Method-Dependent Quality`: While "diversity matters" is known, Figure 6 reveals a non-trivial finding: data preference is orthogonal across pruning methods. A sample that is "Golden" for Wanda can be "Detrimental" for SparseGPT. This context-dependency invalidates universal heuristic filtering.
>    - `Failure of "Averaging Out"`: Standard intuition suggests that larger datasets average out noise. However, our analysis shows that in the highly sensitive regime of pruning calibration, low-quality samples act as "poison." They do not just contribute less; they actively distort the Hessian/activation statistics, degrading the performance of the entire set.

---

> ### Author Response · Authors · 2025-11-27
>
> ### W3. Experiments Coverage
>
> 1. The reviewer claims the experimental section is "quite limited." We respectfully disagree with this claim based on the extensive scope of our evaluation.
>    - Model Coverage: We evaluated 4 distinct models across 2 model families: Llama3.1-8B, Qwen2.5-7B, Llama3.2-3B, and Llama3.2-1B.
>    - Method Coverage: We tested 3 distinct pruning paradigms: Global Structured Pruning (NIRVANA), Local Structured Pruning (LLM-Pruner), and Unstructured Pruning (Wanda).
> 2. The reviewer suggests comparisons with "prior data selection techniques" (e.g., methods for pre-training or instruction tuning). This suggestion is methodologically incompatible because the mathematical objectives are fundamentally disjoint.
>
> To clarify this for the reviewer, we formally distinguish the two problems:
>
> - General Data Selection (Training/SFT): General data selection aims to find a subset $S$ that maximizes performance after gradient updates:
>
> $$\theta_{new} = \theta_{old} - \eta \nabla_{\theta} \mathcal{L}_{train}(S)$$
>
> The goal is to select data that provides the best gradient direction for weight updates.
> - Calibration Data Selection (Pruning): Calibration selection aims to find a subset $C$ that generates the most accurate importance statistics based on the model $f$ to determine a binary mask $M$:
>
> $$M = \text{Top}_k(\text{Saliency}(f, X_C)); $$
>
> $$\theta_{pruned} = \theta_{old} \odot M$$
>
> Here, the model weights $\theta$ are frozen. The data $C$ does not update weights; it determines which weights to keep.
> - Why the Comparison is Invalid: Standard data selection metrics (e.g., training dynamics, forgetting events, gradient diversity) [1,2,3] measure a sample's contribution to learning. They do not measure a sample's contribution to preserving model's original capacity for pruning. Applying a training-centric selector to a zero-shot pruning task is theoretically misaligned. This is why we compared against the only relevant baselines: Entropy (information density) and Synthetic (distribution alignment).
>
>
> [1] Deep Learning on a Data Diet: Finding Important Examples Early in Training, NeurIPS 2021.\
> [2] Understanding Black-box Predictions via Influence Functions, ICML 2017.\
> [3] Submodularity in Data Subset Selection and Active Learning, ICML 2015.
>
>
>
>
> ## Happy to engage in further discussion!
>
> **Thank you again for the thoughtful review. With the valuable feedback from you and the other reviewers, we have put substantial efforts into obtaining new results and expanding our analysis. We hope our responses address your concerns, and we would be happy to discuss further if you have any additional questions.**

---

### Official Review · Reviewer_eJWG · 2025-10-30

**Soundness:** 3
**Presentation:** 2
**Contribution:** 2
**Rating:** 4
**Confidence:** 4

**Summary:**

The OASIS framework proposes a novel approach to improving post-training pruning of large language models (LLMs) by addressing the problem of calibration data selection. Traditional calibration data selection methods rely on simple heuristics, such as random sampling or entropy, which often result in suboptimal and inconsistent pruning outcomes. The authors point out that this inconsistency arises because the importance of calibration samples varies and is context-dependent (i.e., it depends on the specific model and pruning method). A key feature of OASIS is its end-to-end framework, which formulates calibration data selection as an optimization problem and solves it using a differentiable soft-mask proxy. This allows task-level gradients to be backpropagated to the calibration data, dynamically discovering the subset most beneficial for pruning. Experiments show that OASIS improves the performance of various state-of-the-art pruning methods, establishing a new standard for data-aware model compression.

**Strengths:**

1. Context-aware calibration: The adaptive selection of calibration data allows pruning results to be optimized based on the specific model and pruning algorithm, providing high specificity.
2. Improved pruning performance: Compared with traditional heuristic methods, OASIS offers more consistent pruning outcomes and can reduce variance in pruning results.
3. Wide applicability: The method is compatible with various pruning techniques, making it practical and suitable for different types of model compression.

**Weaknesses:**

1. Poor figure readability: The legends and chart sizes in the paper are relatively small. Although the figures are information-dense, it is difficult to extract clear conclusions, which affects readers’ intuitive understanding of the experimental results.
2. Limited improvement for low-accuracy models: When the base model has low accuracy, OASIS provides only minimal gains in perplexity and downstream task performance. For example, for Llama-3.1-8B, the average accuracy is 79.36, which drops to 52.50 after pruning. With OASIS, it only increases to 52.97, indicating that the method does not significantly improve low-accuracy models and does not bring substantial performance breakthroughs.
3. Unclear iterative process and high time cost: OASIS relies on iterative optimization to dynamically select the optimal calibration data subset, but the paper does not specify the number of iterations needed or the computational cost per iteration. This may require significant computational resources and long training time in practical applications, limiting the feasibility of the method.
4. Generality issue: Experiments are conducted only for a 50% pruning rate and models under 8B parameters. It remains unclear how the method performs at higher pruning rates or on larger models, limiting the assessment of its general applicability.

**Questions:**

1. How many iterations are required for the optimization problem to converge, and what is the computational cost per iteration?
2. Does the method still work effectively at lower pruning rates?
3. Can the method be applied to large models, such as Llama-65B, and is the computational cost still acceptable?

---

> ### Author Response · Authors · 2025-11-27
>
> Dear Reviewer eJWG,
>
> We sincerely thank you for your constructive review and insightful questions. Your feedback has been extremely valuable in improving the clarity of our work. We have carefully considered your comments, conducted extensive analyses, and provide detailed responses below.
>
> ### W1. Readability of Figures
>
> We thank the reviewer for pointing this out. We have revised the paper to enlarge the font sizes of legends and axis labels and have optimized the chart layout to remove excess whitespace. We ensure that all figures in the final version will be clear and information-dense without sacrificing readability.
>
> ### W2 & Q2. Performance Improvement
>
>
> 1. **Magnitude of Improvement:** The reviewer noted that the absolute accuracy gain on Llama-3.1-8B seems limited. We respectfully emphasize that this configuration corresponds to a very aggressive 50% *structured* pruning regime, where even strong baselines in prior work suffer substantial degradation; this is a general limitation of current pruning techniques rather than a specific issue of OASIS. In such a challenging regime, our goal is not to "create" large absolute gains, but to **stabilize pruning by avoiding bad calibration batches**. Across all models (Llama-1B/3B/8B, Qwen-7B) and all pruning methods (NIRVANA, LLM-Pruner, Wanda), OASIS consistently improves over their respective baselines. This indicates that OASIS effectively reduces variance and failure cases caused by suboptimal calibration data, even when the overall task is intrinsically difficult.
> 2. Effectiveness at Lower Sparsity Rates: The reviewer also asked if the method works at lower pruning rates. We actually provided evidence for this in the original manuscript. As noted in Table 1, for Qwen2.5-7B with LLM-Pruner, we set the sparsity to 40% (lower than the standard 50%) due to the model's sensitivity. Even at this lower sparsity (40%), OASIS remained effective and improved the downstream performance. This confirms that OASIS remains highly effective at identifying critical data even when the pruning rate is lower (40%), and we expect similar benefits at 20–30% sparsity.
>
> ### W3 & Q1. Time Cost
>
> The reviewer asked about the number of iterations and computational cost.
>
> - Iterations: OASIS typically converges very quickly. In our experiments, we trained the data weights for only 3 epochs.
> - Time Cost: As also suggested by Reviewer 6k7x, we have documented the exact wall-clock time required for OASIS to complete these 3 epochs on a single NVIDIA GH200 GPU:
>
> | Model       | Parameters | Time Cost |
> | ----------- | ---------- | --------- |
> | Llama3.2-3B | 3B         | 46mins    |
> | Qwen2.5-7B  | 7B         | 73mins    |
> | Llama3.1-8B | 8B         | 87mins    |
>
> This demonstrates that OASIS is a lightweight pre-processing step (taking roughly 1 hour) compared to the significant cost of training or recovering a model.
>
> ### W4 & Q3. Generalizability
>
> The reviewer asked about scaling to larger models.
>
> - Trend Verification: We have validated OASIS across a wide range of sizes: 1B $\to$ 3B $\to$ 7B $\to$ 8B. The performance improvement is consistent across this order-of-magnitude scaling, which strongly suggests the method generalizes well to larger models.
> - Academic Constraints: 7B/8B is currently the standard scale for academic pruning research (e.g., SparseGPT, Wanda). Running iterative optimization on 65B+ models requires industrial-level compute clusters (multiple A100/H100 nodes for model parallelism) that are unfortunately beyond the resource constraints of our academic lab. However, since OASIS optimization cost scales linearly with the forward pass cost, it remains theoretically feasible for industrial deployment.
>
> ## Happy to engage in further discussion!
>
> **Thank you again for the thoughtful review. With the valuable feedback from you and the other reviewers, we have put substantial efforts into obtaining new results and expanding our analysis. We hope our responses address your concerns, and we would be happy to discuss further if you have any additional questions.**

---

### Official Review · Reviewer_6k7x · 2025-10-31

**Soundness:** 4
**Presentation:** 4
**Contribution:** 2
**Rating:** 4
**Confidence:** 4

**Summary:**

The paper identifies that pruning large language models is highly sensitive to the calibration data used, and that existing heuristic-based methods often lead to inconsistent and suboptimal results due to data quality variance. To address this, it proposes OASIS, a fully differentiable framework that optimizes calibration data selection end-to-end by backpropagating task-level gradients through a soft-mask proxy, allowing the model to learn which samples most improve post-pruning performance. Experiments on structured and unstructured pruning across Llama and Qwen models show that OASIS consistently outperforms heuristic and synthetic data baselines, establishing a new standard for data-aware model compression.

**Strengths:**

1. This paper provides a thorough investigation of the impact of calibration data on pruning from both macro and micro perspectives, offering valuable insights for future research in this area.

2. The experiments are solid and comprehensive, covering multiple LLMs under both structured and unstructured pruning settings, which strongly support the paper’s conclusions.

3. The writing is well-organized and easy to follow.

**Weaknesses:**

1. The macro-level conclusions have already been established in prior work, so the novelty in this aspect appears limited.

2. The motivation for introducing noise perturbations into the input is not clearly explained. Although the authors demonstrate its effectiveness through ablation studies, it remains unclear why adding noise would lead to more stable optimization.

3. The paper does not report the time or computational cost of data selection. Excessive overhead could undermine the practical value of the proposed method. If I allocate the same computational cost for data selection to gradient-based iterative pruning or recovery training, would it yield better performance?

**Questions:**

1. Why would adding noise lead to more stable optimization?

2. If I allocate the same computational cost for data selection to gradient-based iterative pruning or recovery training, would it yield better performance?

---

> ### Author Response · Authors · 2025-11-27
>
> Dear Reviewer 6k7x,
>
> We sincerely thank you for your constructive review and insightful questions. Your feedback has been extremely valuable in improving the clarity of our work. We have carefully considered your comments, conducted extensive analyses, and provide detailed responses below.
>
> ### W1: Novelty
>
> **Novelty: Context-Dependency & Counter-Intuitive Poisoning**: The reviewer correctly notes that the general importance of calibration data (macro-level) has been touched upon in prior works. However, our work **fundamentally advances** the field by identifying why heuristic selection fails and offering a solution for context-dependent selection.
>    - `Method-Dependent Quality`: While "diversity matters" is known, Figure 6 reveals a non-trivial finding: data preference is orthogonal across pruning methods. A sample that is "Golden" for Wanda can be "Detrimental" for SparseGPT. This context-dependency invalidates universal heuristic filtering.
>    - `Failure of "Averaging Out"`: Standard intuition suggests that larger datasets average out noise. However, our analysis shows that in the highly sensitive regime of pruning calibration, low-quality samples act as "poison." They do not just contribute less; they actively distort the Hessian/activation statistics, degrading the performance of the entire set.
>
> ### W2 & Q1: Effect of Noise
>
>
> We thank the reviewer for this insightful request. In the original manuscript, we included Figure 5 to visually demonstrate how noise stabilizes the distribution of data weights during optimization. However, we agree that providing the final quantitative performance is necessary to fully justify this design choice.
>
> 1. **Theoretical Justification**: The necessity of perturbation stems from the high-dimensional nature of the optimization landscape.
>    - `Escaping Local Optima`: The optimization of data weights $u$ w.r.t. the task loss is non-convex and high-dimensional. Without perturbation, the gradient descent process is prone to getting trapped in sharp, narrow local minima (local optima), effectively overfitting to spurious correlations in specific tokens
>    - `Manifold Regularization`: Adding noise to the embeddings effectively smooths the loss landscape. As established in classical learning theory [1], training with noise is mathematically equivalent to Tikhonov regularization, which penalizes the gradient norm and favors flatter, more robust solutions.
>    - `Consistency in NLP`: This technique has been successfully adapted to NLP to improve generalization and robustness in methods like SMART [2] and FreeLB [3]. In OASIS, this "smoothing" ensures that the selected data weights reflect true signal importance rather than noise in the calibration batch.We have added this table and the expanded theoretical discussion to the revised paper.
> 2. **Quantitative Impact of Noise**: As suggested by Reviewer w7F5, we evaluated the final pruned models with and without the embedding perturbation (noise) during the OASIS selection phase. The results are presented below:
>
>
> | Method                         | WikiT ↓    | PTB ↓      | LambD  ↓   |
> | ------------------------------ | ---------- | ---------- | ---------- |
> | OASIS (NIRVANA+Llama3.1-8B)    | **36.37**  | **52.92**  | **52.92**  |
> | w/o noise                      | 38.65      | 58.91      | 61.25      |
> | OASIS (LLM-Pruner+Llama3.1-8B) | **173.51** | **356.02** | **209.29** |
> | w/o noise                      | 178.47     | 366.02     | 212.84     |
> | OASIS (Wanda+Llama3.1-8B)      | 9.34       | **14.47**  | 23.48      |
> | w/o noise                      | 9.34       | 14.56      | 23.48      |
>
> Observations:
>
> - `Structured Pruning is More Sensitive`: For NIRVANA and LLM-Pruner, removing the noise leads to a noticeable degradation in perplexity. This aligns with our finding that structured pruning is highly sensitive to calibration data quality; without noise, the selection overfits to suboptimal samples.
> - `Unstructured Pruning is More Stable`: For Wanda, the impact is less pronounced. This is consistent with the generally robust nature of weight-level magnitude pruning, which is less affected by slight shifts in calibration statistics.
>
>
> [1] Training with Noise is Equivalent to Tikhonov Regularization, Neural Computation 1995.\
> [2] SMART: Robust and Efficient Fine-Tuning for Pre-trained Natural Language Models through Principled Regularized Optimization, ACL 2020.\
> [3] FreeLB: Enhanced Adversarial Training for Natural Language Understanding, ICLR 2020.

---

> ### Author Response · Authors · 2025-11-27
>
> ### W3 & Q2: Costs of OASIS
>
> A full compute-parity comparison with recovery training is an interesting direction but outside the scope / resource of this submission, because the computational scales of these two tasks are fundamentally different (orders of magnitude apart).
>
> - OASIS: We optimize a tiny vector of scalar weights $N$ while keeping the LLM backbone completely frozen. This is a lightweight optimization problem.
> - Recovery Training: This involves updating billions of parameters (e.g., 8B for Llama3.1). Even a short fine-tuning session requires massive compute to compute gradients for all weights.
> - Specific Time Cost: Per your request, we have documented the exact wall-clock time required for OASIS to converge on NVIDIA GH200 GPUs:
>
> | Model       | Parameters | Time Cost |
> | ----------- | ---------- | --------- |
> | Llama3.2-3B | 3B         | 46mins    |
> | Qwen2.5-7B  | 7B         | 73mins    |
> | Llama3.1-8B | 8B         | 87mins    |
> | Llama3.1-8B (Recovery Fine-tuning w/ LoRA)| 8B         | >4hrs    |
>
> Comparison Analysis:
>
> - Vs. Recovery Fine-tuning: Pruning with suboptimal data causes structural damage to the model. Allocating ~1.5 hours to recovery training is often insufficient to fully repair this damage, as the model may have lost critical capabilities that are hard to recover in a short fine-tuning window.
> - Vs. Iterative Mask Search: Compared to other advanced pruning strategies that search for optimal masks (e.g., methods like Týr-the-Pruner [1] or similar iterative approaches which can take 10+ hours to search), OASIS is significantly more efficient.
> - Value Proposition: OASIS ensures the pruning process itself preserves the maximum amount of information. If one chooses to perform recovery training, OASIS provides a superior initialization with significantly lower perplexity. This ensures that the expensive recovery training is spent on improving a high-quality model rather than repairing a broken one, making OASIS complementary and highly efficient.
>
>
> [1] Týr-the-Pruner: Structural Pruning LLMs via Global Sparsity Distribution Optimization, NeurIPS 2025.
>
> ## Happy to engage in further discussion!
>
> **Thank you again for the thoughtful review. With the valuable feedback from you and the other reviewers, we have put substantial efforts into obtaining new results and expanding our analysis. We hope our responses address your concerns, and we would be happy to discuss further if you have any additional questions.**

---

### Official Review · Reviewer_w7F5 · 2025-11-01

**Soundness:** 2
**Presentation:** 3
**Contribution:** 2
**Rating:** 4
**Confidence:** 3

**Summary:**

This paper proposes a data selection method for LLM pruning. The authors first investigate how different data selection strategies affect pruning performance from both macro and micro perspectives, and find that "heuristics fail in calibration data selection." They then propose an algorithm called OASIS to select datasets for pruning tasks. Experiments demonstrate that OASIS outperforms other data selection approaches and is suitable for both structured and unstructured pruning.

**Strengths:**

1. A comprehensive study on how various calibrated data selection strategies affect model pruning performance, encompassing both structured and unstructured pruning methods.

2. The proposed algorithm is straightforward and easy to implement, while consistently improving upon the performance of baseline methods in experiments.

**Weaknesses:**

1. The findings are sound but not very surprising. First, it is apparent that performance saturates as data size increases, while data diversity significantly impacts model performance including in model pruning, and the optimal data composition varies across different tasks. Additionally, the statement "A single low-quality ('detrimental') sample can contaminate the entire set and severely degrade the performance of a high-quality ('golden') set" is somewhat confusing. What exactly is the size of the "entire set"? For example, if we have a selected calibrated set chosen by OASIS and introduce just one low-quality sample, will the performance indeed degrade severely?

2. The perturbation of embeddings requires further justification. Specifically, it is unclear why such perturbation ensures stability. Moreover, the ablation studies do not report the final model performance without perturbation.

**Questions:**

1. It would be helpful to include a small experiment showing how performance degrades when a single detrimental sample is added to a high-quality calibrated set of realistic size.

2. Ablation study should report final model performance without perturbation to better illustrate its contribution.

3. It appears that the reported code site has no content.

---

> ### Author Response · Authors · 2025-11-27
>
> Dear Reviewer w7F5,
>
> We sincerely thank you for your constructive review and insightful questions. Your feedback has been extremely valuable in improving the clarity of our work. We have carefully considered your comments, conducted extensive analyses, and provide detailed responses below.
>
> ### W1 & Q1: Effect of Detrimental Samples
>
> 1. **Novelty: Context-Dependency & Counter-Intuitive Poisoning**: We thank the reviewer for recognizing the effectiveness of OASIS. Regarding the "surprising" nature of our findings, we respectfully highlight two key insights that distinguish our work from general data selection wisdom:
>    - `Method-Dependent Quality`: While "diversity matters" is known, Figure 6 reveals a non-trivial finding: data preference is orthogonal across pruning methods. A sample that is "Golden" for Wanda can be "Detrimental" for SparseGPT. This context-dependency invalidates universal heuristic filtering.
>    - `Failure of "Averaging Out"`: Standard intuition suggests that larger datasets average out noise. However, our analysis shows that in the highly sensitive regime of pruning calibration, low-quality samples act as "poison." They do not just contribute less; they actively distort the Hessian/activation statistics, degrading the performance of the entire set.
> 2. **Experiment: The Impact of Detrimental Samples (Poisoning Effect)**: The reviewer asked a critical question: Does adding a single detrimental sample to a realistic-sized high-quality set actually degrade performance? Per your suggestion, we conducted this experiment. We took the optimized calibration set selected by OASIS ($N=32$) and deliberately injected a single "Detrimental" samples identified in our micro-analysis.
>
> | Method                                   | WikiT ↓    | PTB ↓      | LambD  ↓   | Avg        |
> | ---------------------------------------- | ---------- | ---------- | ---------- | ---------- |
> | Golden set (NIRVANA+Llama3.2-1B)         | **179.02** | **202.86** | **182.65** | **188.18** |
> | Golden set + a single detrimental sample | 196.62     | 378.99     | 184.70     | 253.44     |
> | Golden set (Wanda+Llama3.2-1B)           | **21.05**  | **38.11**  | **50.50**  | **36.76**  |
> | Golden set + a single detrimental sample | 22.24      | 39.21      | 52.27      | 37.91      |
>
> This result empirically confirms that the inclusion of a few "poisonous" samples can undo the benefits of the optimized set. This confirms that exclusion is just as important as selection, which random sampling cannot guarantee.

---

> ### Author Response · Authors · 2025-11-27
>
> ### W2 & Q2: Effect of Noise
>
> We thank the reviewer for this insightful request. In the original manuscript, we included Figure 5 to visually demonstrate how noise stabilizes the distribution of data weights during optimization. However, we agree that providing the final quantitative performance is necessary to fully justify this design choice.
>
> 1. **Quantitative Impact of Noise**: Per your suggestion, we evaluated the final pruned models with and without the embedding perturbation (noise) during the OASIS selection phase. The results are presented below:
>
>
> | Method                         | WikiT ↓    | PTB ↓      | LambD  ↓   |
> | ------------------------------ | ---------- | ---------- | ---------- |
> | OASIS (NIRVANA+Llama3.1-8B)    | **36.37**  | **52.92**  | **52.92**  |
> | w/o noise                      | 38.65      | 58.91      | 61.25      |
> | OASIS (LLM-Pruner+Llama3.1-8B) | **173.51** | **356.02** | **209.29** |
> | w/o noise                      | 178.47     | 366.02     | 212.84     |
> | OASIS (Wanda+Llama3.1-8B)      | 9.34       | **14.47**  | 23.48      |
> | w/o noise                      | 9.34       | 14.56      | 23.48      |
>
> Observations:
>
> - `Structured Pruning is More Sensitive`: For NIRVANA and LLM-Pruner, removing the noise leads to a noticeable degradation in perplexity. This aligns with our finding that structured pruning is highly sensitive to calibration data quality; without noise, the selection overfits to suboptimal samples.
> - `Unstructured Pruning is More Stable`: For Wanda, the impact is less pronounced. This is consistent with the generally robust nature of weight-level magnitude pruning, which is less affected by slight shifts in calibration statistics.
>
> 2. **Theoretical Justification**: The necessity of perturbation stems from the high-dimensional nature of the optimization landscape.
>    - Escaping Local Optima: The optimization of data weights $u$ w.r.t. the task loss is non-convex and high-dimensional. Without perturbation, the gradient descent process is prone to getting trapped in sharp, narrow local minima (local optima), effectively overfitting to spurious correlations in specific tokens
>    - Manifold Regularization: Adding noise to the embeddings effectively smooths the loss landscape. As established in classical learning theory [1], training with noise is mathematically equivalent to Tikhonov regularization, which penalizes the gradient norm and favors flatter, more robust solutions.
>    - Consistency in NLP: This technique has been successfully adapted to NLP to improve generalization and robustness in methods like SMART [2] and FreeLB [3]. In OASIS, this "smoothing" ensures that the selected data weights reflect true signal importance rather than noise in the calibration batch.We have added this table and the expanded theoretical discussion to the revised paper.
>
> [1] Training with Noise is Equivalent to Tikhonov Regularization, Neural Computation 1995.\
> [2] SMART: Robust and Efficient Fine-Tuning for Pre-trained Natural Language Models through Principled Regularized Optimization, ACL 2020.\
> [3] FreeLB: Enhanced Adversarial Training for Natural Language Understanding, ICLR 2020.
>
>
> ### Q3: Code Availability
>
> We have double-checked the anonymous repository link https://anonymous.4open.science/r/OASIS-DC05/ provided in the submission, Page 10, Line 506-507, and confirmed it is accessible. We are happy to provide it again or troubleshoot if there are access issues.
>
> ## Happy to engage in further discussion!
>
> **Thank you again for the thoughtful review. With the valuable feedback from you and the other reviewers, we have put substantial efforts into obtaining new results and expanding our analysis. We hope our responses address your concerns, and we would be happy to discuss further if you have any additional questions.**

---

### Note · Authors · 2025-12-30

I have read and agree with the venue's withdrawal policy on behalf of myself and my co-authors.